# Development and Evaluation of a Mobile Application with Augmented Reality for Guiding Visitors on Hiking Trails

**Rute Silva** [1], **Rui Jesus** [2,*] **and Pedro Jorge** [2]

1    ISEL—Instituto Superior de Engenharia de Lisboa/IPL, 1959-007 Lisbon, Portugal; a42415@alunos.isel.pt
2    Lisbon School of Engineering (ISEL)/IPL and NOVA LINCS, 1959-007 Lisbon, Portugal;
     pedro.mendes.jorge@isel.pt
*    Correspondence: rui.jesus@isel.pt

**Abstract:** Tourism on the island of Santa Maria, Azores, has been increasing due to its characteristics in terms of biodiversity and geodiversity. This island has several hiking trails; the available information can be consulted in pamphlets and physical placards, whose maintenance and updating is difficult and expensive. Thus, the need to improve the visitors' experience arises, in this case, by using the technological means currently available to everyone: a smartphone. This paper describes the development and evaluation of the user experience of a mobile application for guiding visitors on said hiking trails, as well as the design principles and main issues observed during this process. The application is based on an augmented reality interaction model providing visitors with an interactive and recreational experience through Augmented Reality in outdoor environments (without additional marks in the physical space and using georeferenced information), helping in navigation during the route and providing updated information with easy maintenance. For the design and evaluation of the application, two studies were carried out with users on-site (Santa Maria, Azores). The first had 77 participants, to analyze users and define the application's characteristics, and the second had 10 participants to evaluate the user experience. The feedback from participants was obtained through questionnaires. In these questionnaires, an average SUS (System Usability Scale) score of 83 (excellent) and positive results in the UEQ (User Experience Questionnaire) were obtained.

**Keywords:** mobile augmented reality; mobile user experience; mobile user interface; location-based applications

## 1. Introduction

Santa Maria is the oldest island of the Azores archipelago; therefore, its origin and history generate great curiosity in the scientific community because of its biodiversity and geodiversity. The island is located to the south and east of the archipelago, integrating the Eastern Group, and its drier and warmer climate makes it known as the Island of the Sun.

Santa Maria Natural Park, as shown in Figure 1, is composed of different areas that are classified into two Natural Reserves, one Natural Monument, four Habitat/Species' Management Areas, three Protected Landscape Areas, and three Protected Areas with Sustainable Use of Natural Resources. To better explore the island, there are six official trails and one great route that goes through some of the Natural Park areas, these being: *PR01SMA Costa Norte*, *PRC02SMA Pico Alto*, *PRC03SMA Entre a Serra e o Mar*, *PR04SMA Santo Espírito-Maia*, *PR05SMA Costa Sul*, *PR06SMA Areia Branca* and *GR01SMA Grande Rota de Santa Maria*.

Currently, most of the information about the trails is available on posters and pamphlets (physical and online). These information posters are generally distributed at the beginning of the trails, but also near the main points of interest along each of them. The placards are often difficult to read and easily damaged, both by the sun and humidity and by people. Although these posters are easily accessible and no other means is required to access that information, they are not easy to maintain or update.

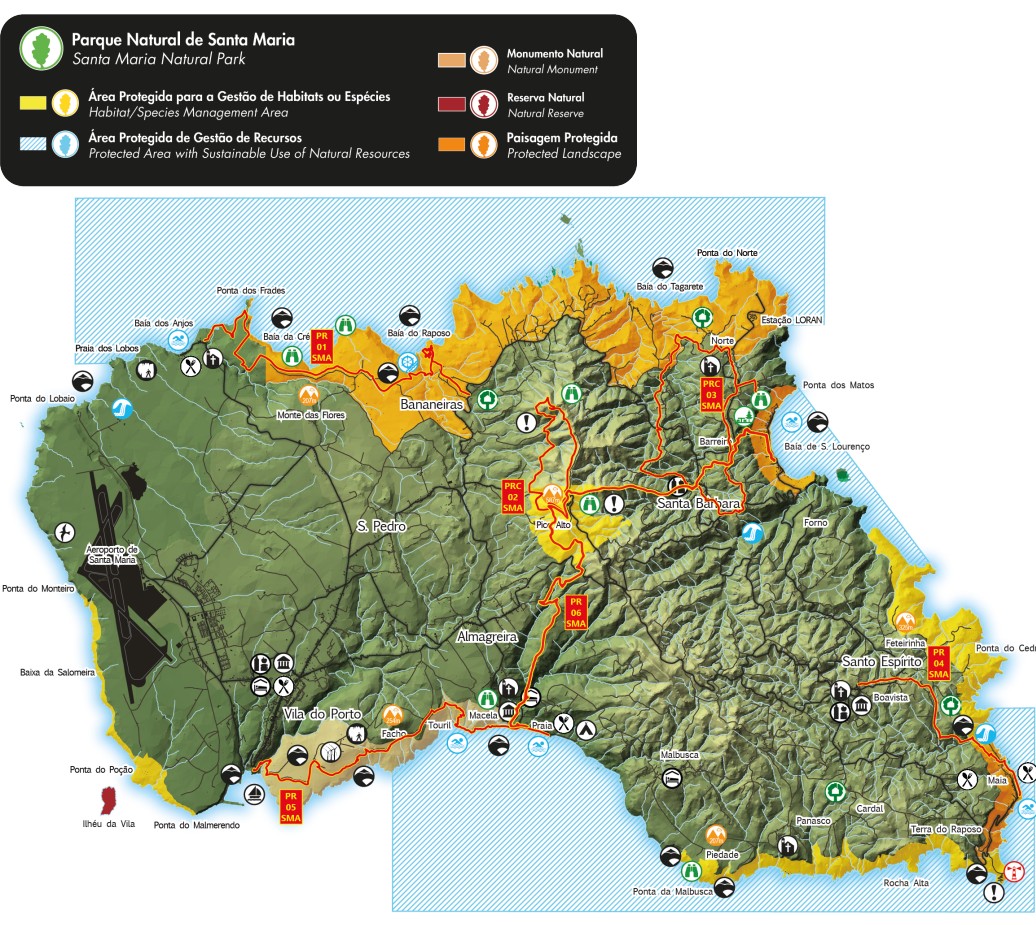

**Figure 1.** Santa Maria Natural Park (Adapted figure from the pamphlet of *GR01SMA Grande Rota de Santa Maria*).

The main goal of this work [1] is to develop and evaluate an application for mobile devices that provides the user with a more interactive and ludic experience through the use of Augmented Reality (AR), navigation assistance for visitors, and access to information that is easy to maintain and update. Thus, it is intended to expand and attract even more locals and visitors to the protected areas of the island and its specificities, as well as to new technologies and the various possibilities that they provide.

The use of Augmented Reality to enhance the interactive experience with the physical space represents the added value of this work, as it generates and increases curiosity about the points of interest of the island and about this technology, contributing to providing a richer experience. This paper describes the study conducted to develop and evaluate a mobile application to enrich the user experience by guiding visitors on hiking trails. The study is focused on the use of augmented reality technology as a way to interact with the environment.

The paper is structured as follows: Section 2 discusses the related work; Section 3 presents the analysis and methodology used; Section 4 describes the development process; Section 5 presents the evaluation of the mobile application and discusses its results; Section 6 presents the conclusions and future work.

## 2. Related Work

The study described in this paper is related to works and approaches developed in different scopes like Navigation, Mobile Tour Guides, or Augmented Reality in tourism.

## 2.1. Navigation

In general, mobile applications associated with hiking trails need to use concepts in the navigation field [2] related to localization and orientation. To determine positions and distances, usually, they used Global Navigation Satellite System (GNSS) signals and the Haversine equation [3,4]. The diagram of Figure 2 helps understand the terms used, such as heading, true bearing, and relative bearing [5,6].

The horizontal angle between the magnetic North and the direction of the device is called **heading**, and this is the angle indicated by the compass. **True bearing** represents the horizontal angle between the North and the line that links the device to a point of interest. **Relative bearing** represents the horizontal angle between the heading and the true bearing.

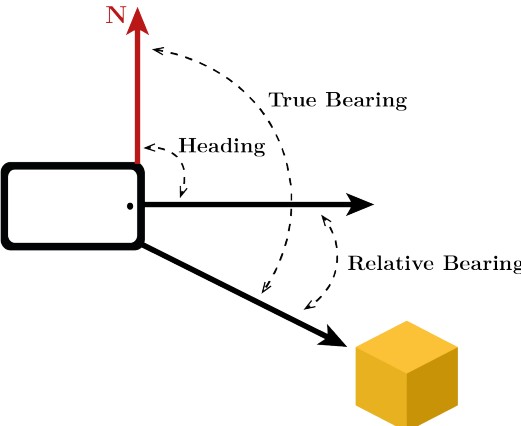

**Figure 2.** Representation of the concept used in orientation.

## 2.2. Mobile Tour Guides

The audio guides were the first ways to introduce multimedia content to the visitors' experience [7,8]. Mobile tour guides replaced audio guides with the introduction of new mobile technologies in tourism. In general, these mobile tour guides should provide, fully or partially, four of these functionalities: navigation services, content-based services, communication/social services, and commercial services [9].

Some studies have been carried out on the acceptance of mobile tourist guides by visitors [10–12], demonstrating that the main factor that leads visitors to use this type of guide is the quality of the information available, as well as the type of experience they provide. In addition, they bring a degree of independence and flexibility to users that many traditional guides do not offer.

Regarding the usability of this type of tour guide applications, in [13] a study is presented where it is possible to observe that the main measures of user experience in this type of scenario are: the quality of interaction between users and mobile guides (these should not be a barrier, but an extension); the ease of learning how to use and control the mobile guide; and overall usability (through the ability to convey more information about the points of interest/exhibitions and the enrichment of their experience).

The work published in [14] presents an exhaustive study of various mobile tour guides, ranking them in terms of their development approach, availability, customization, usability, social aspect, and the type of evaluation used.

In [15], a mobile guide is proposed for cultural heritage sites based on pictures. It presents information related to the place based on pictures and includes sharing features that help the visitor understand the place. One pioneer work including games in cultural heritage was proposed by Correia et al. [16]. The goal of this work is to define and implement a platform for mobile storytelling, information access, and gaming activities that are evaluated in a cultural heritage site.

These mobile applications and other digital platforms are expected to enhance and improve the engagement and satisfaction of the tourists exploring cultural and heritage

sites. Therefore, [17] presents a location-aware mobile application that supports users' exploration of sites of heritage and cultural value in Singapore, facilitating better navigation and digital socializing.

### 2.3. Augmented Reality

Milgram and Kishino [18] describe the notion of a virtuality continuum (Figure 3), where at one end are the totally real environments, and at the other end are the totally virtual environments (Virtual Reality). Mixed Reality is everything in between these two extremes and is the most general and broad term to describe the core area of the virtuality continuum.

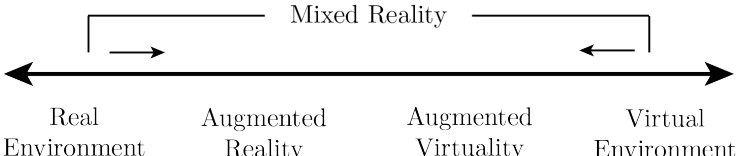

**Figure 3.** Simplified representation of *virtuality continuum* (based on the figure proposed by Milgram and Kishino [18]).

To better define this gray area, two concepts were adopted that are distinguished by the "amount" of real environment present, that is, if the environment is predominantly real or virtual. Thus, in Augmented Reality, the real world is enhanced by virtual objects, and in Augmented Virtuality, real objects are integrated into the virtual environment. Ronald Azuma [19] defines Augmented Reality as a system that combines real and virtual, is interactive in real-time, and registered in 3D. In this study, we follow this definition of AR.

One of the main challenges of AR is motion tracking and geo-localization [20]. The success of this technology is based on the accuracy with which it is possible to establish the position and orientation of the camera in a three-dimensional space, the so-called pose. To do this, sensors such as accelerometer, gyroscope, compass, camera or GNSS data can be used to establish the pose with the best accuracy possible.

Depending on its application, there are several types of AR [20–22]. It can be marker-based, by using fiducial marks or Natural Feature Tracking (NFT), or markerless, by using Simultaneous Localization and Mapping (SLAM).

The virtual object is placed in the scene according to its distance from the camera and with a certain rotation on itself. The calculation of this rotation is important to create an even more realistic experience for the user, since different views of the virtual object must be rendered depending on the user's orientation.

Different solutions have been proposed for tourism using Augmented Reality [23,24]. Initially, the main problems for the widespread implementation of this type of technology were the lack of funding for equipment acquisition [25], the interoperability between the platforms available for the development of this type of AR applications [26], and the hardware resources needed [27].

A pioneering study by Aluri [28] investigated an existing mobile reality application with a gaming purpose (Pokemón Go) and examined the intentions to use it as a mobile travel guide in the future.

In [29], the development of a guide with smart glasses is proposed that provides visitors with an outdoor museum with context-aware information regarding the points of interest in their field of view (using orientation and location functionalities). This smart glass-based guide is also compared with an identical smartphone-based application, concluding that the smart glasses offer a more seamless and private experience, while the smartphone offers a better group experience, due to the possibility of information sharing in its exposed display.

More recently, Grammatikopoulou and Grammalidis [30] propose a self-guided tour tool with augmented reality and social features that enables information sharing and interactions between museum staff and visitors to create an online learning community.

Mobile platforms using Augmented Reality aim to provide users with a more fluid experience in which the obstruction of space by virtual objects is not evident and too intrusive [31] and for that, an approach focused on the user and the purpose of the platform is often used [32,33].

Some existing and published mobile applications, such as SmartGuide [34], Pocket-Sights [35], and AR Trails [36], have similar objectives and functionalities as the proposed application. Our proposal is focused on the study of using an interaction model based on augmented reality to improve the user experience.

## 3. User Research

The work described in this paper proposes a way to explore relevant information in outdoor environments, making it possible to provide additional information that is easy to maintain and update. The main idea is to help the user navigate and observe the physical space using several sensors of the mobile device (including geolocation) and presenting information (including 3D content) in the form of Augmented Reality to enrich the user experience.

The domain of our study is the hiking trails on the island of Santa Maria. We started our study by performing a user research questionnaire to 77 users including residents and visitors of the island.

### 3.1. Analysis

In order to characterize the target audience and understand their needs, a questionnaire was conducted which counted 77 responses from residents (63–81.8%) and visitors (14–18.2%) of the island of Santa Maria. Of these, only 12 had never been on a trail, and the most popular trail was the *PRC02SMA Pico Alto* (completed by 51 respondents), followed by the *PRC03SMA Entre a Serra e o Mar* and *PR05SMA Costa Sul* trails (completed by 40 and 39 respondents, respectively), as shown in the graph in Figure 4.

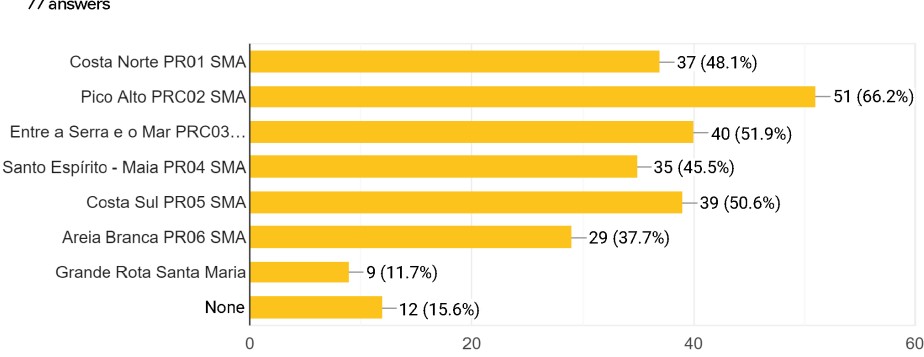

**Figure 4.** Responses to the question "What trails have you completed?"

When asked about the frequency with which they hike, 40.3% reveal that they do it very rarely (about once a year), 26% rarely (about once a month), and 20.8% regularly (about once a week). According to the graph of Figure 5, the major motivation for going on the trails is the connection with nature that they provide (76.6%); however, they also consider that it is important for them to know more about the history of the island and that this is a good way to spend time with friends (40.3% in both cases).

What is your motivation for going on the hiking trails?
77 answers

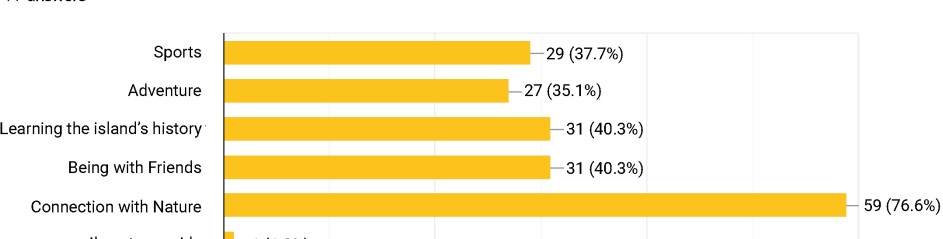

**Figure 5.** Responses to the question "What is your motivation for going on the hiking trails?"

About 70% of those questioned consider that they do not have enough information about the island's points of interest, and this is also the biggest difficulty revealed (pointed out by 41 of those questioned), as well as the orientation during the tours (pointed out by 29 respondents).

All the participants consider a mobile application to guide the hiking trails in Santa Maria useful, and 91% of them "agree" or "completely agree" that they would frequently use this application.

A word cloud (words in the Portuguese language) was created to illustrate the main points of interest mentioned by the participants (Figure 6). In general, the participants think that all ("Todos") the points of interest are important, but the most mentioned ones are "Gruta do Figueiral", "Barreiro da Faneca", and "Costa Sul".

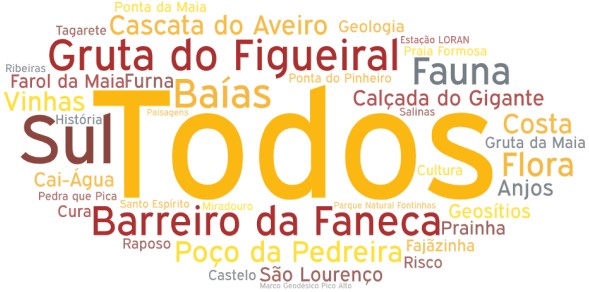

**Figure 6.** Word cloud of points of interest.

When asked about the technologies they would like to see in an application of this type, most mentioned "Navigation during the route (via GPS)" (62 respondents). Text and audio information about points of interest were mentioned by 46 and 30 participants, respectively, and "Augmented Reality" only by 16 of them. More users were expected to mention "Augmented Reality", but this can be justified by some lack of knowledge of the technology. In a study presented later in this document, participants were curious about using this technology and mentioned that they enjoyed the experience.

*3.2. Requirements and Use Cases*

Based on the user research, the user tasks for the application were defined for two different roles:

- User:
  - Make the registration and authentication using three ways: (1) Anonymous, (2) Email and Password, and (3) Google account;
  - View information about the trails: difficulty, duration, length, map, topography, main points of interest, and images;

– Start the navigation on a trail (with information about the distance traveled and time elapsed since the beginning of the trail);
– Enter the Augmented Reality mode when you are close enough to a point of interest;
– Save the completed trail (with information on distance traveled, duration, and points of interest visited along the way);
– View the previously completed trails;
– Delete previously completed trails.
• Administrator
– Add points of interest to available trails;
– Delete points of interest from the available trails.

## 4. Development of the Mobile Application

To implement the system, in the first phase, the concept phase, a study of the technologies available to be used is performed to select the more suitable technologies to use. In the second phase, the construction phase, the architecture of the system is translated into Dart/Flutter code and the application is fully implemented.

### 4.1. Functional Architecture

To meet these requirements, a system based on a client-server architecture (including services in external clouds) was implemented.

This system consists of a mobile application (**Client**) that includes two main modules (**Navigation** and **Augmented Reality**) with the functionalities defined previously (Section 3.2), depending on the type of user. To do this, it is necessary to access the **Sensors** available on the smartphone, as well as a set of tools available to implement the Augmented Reality component.

The user will perform the Authentication on the mobile phone but all the logic process is conducted in the backend (**Server** and **Database**), which also includes the storage of multimedia information (text, images, and virtual objects). The diagram in Figure 7 illustrates the architecture of the application.

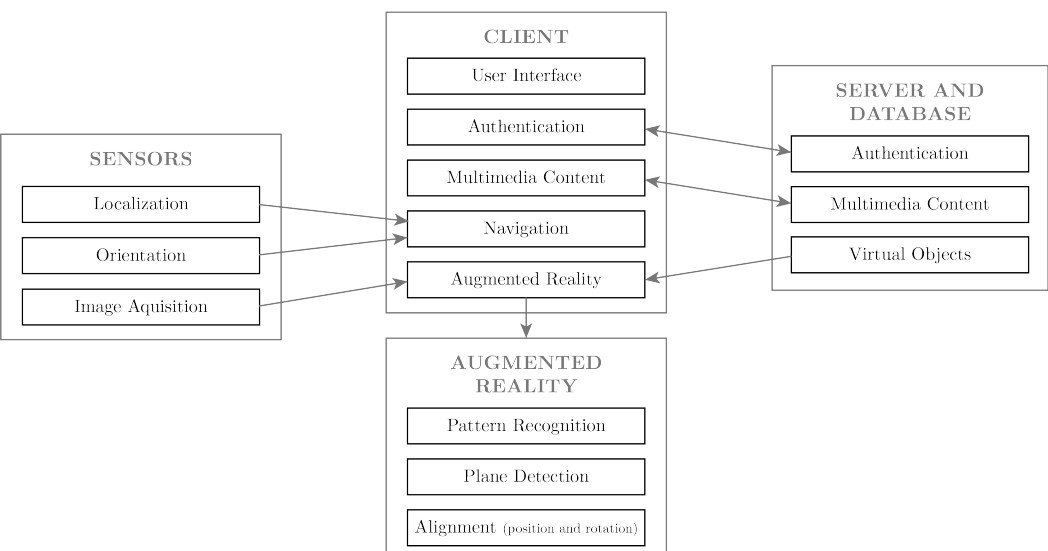

**Figure 7.** Functional architecture of the mobile application.

The application is implemented following a 3-Layer Logical Architecture as the architecture pattern, where the subsystems are divided into three layers: Presentation, Domain, and Data Access (Figure 8). The mobile application consists of seven main screens, each one

being responsible for one main functionality. The following sections explain the principal screens as well as the related logic.

**Figure 8.** 3-Layer Logical Architecture divided by subsystems: Presentation (screens), Domain (domain), and Data Access (data).

*4.2. Technological Architecture*

In the process of choosing the technologies, research was conducted in order to understand which are the most widely used technologies and which best meet the objectives of this project.

Three leading technologies/frameworks were analyzed for the development of the mobile application: Ionic, React Native, and Flutter. The first two are the oldest and best known and the last one is a newer technology with higher popularity [37,38].

Flutter was chosen for the development of the application since it is multiplatform and has proven to be a powerful technology with advantages for this type of project, including the fact that it is natively compiled, increasing its performance when compared to other frameworks [39].

The application developed in this work has some degree of relationship between the data, but implementing a relational database in such a project is not justified. Therefore, it was decided to use Firebase, a Backend as a Service (BaaS) platform from Google, for the development of mobile and web applications, which, among other services, provides authentication (Firebase Authentication), a non-relational database that allows you to store a larger amount of data and make it available in real-time to all users (Firebase Cloud Firestore), and a cloud storage service can store images, videos, and audio, among other types of files (Firebase Cloud Storage).

Although Firebase has a cloud storage service, it is based on absolute paths, which makes it very difficult to access and handle the virtual objects on this platform. Thus, GitHub was used as a Content Delivery Network (CDN), i.e., a distribution network that enables faster content delivery to a larger number of users. In addition, GitHub is based on relative paths, a fundamental aspect in the handling of the referenced files in the virtual objects. The virtual object were modeled and manipulated using Blender.

The selection of Flutter for the development of the mobile application narrowed down the choice of platform to integrate Augmented Reality. Thus, ARCore was chosen, since it is multiplatform and there is already a compatible plugin developed for integration with Flutter. However, the documentation regarding this plugin is limited, and not all the features provided by ARCore are implemented, which limited the development of some of the functionalities of this work.

Figure 9 shows the architecture of the application with the associated technological dependencies chosen.

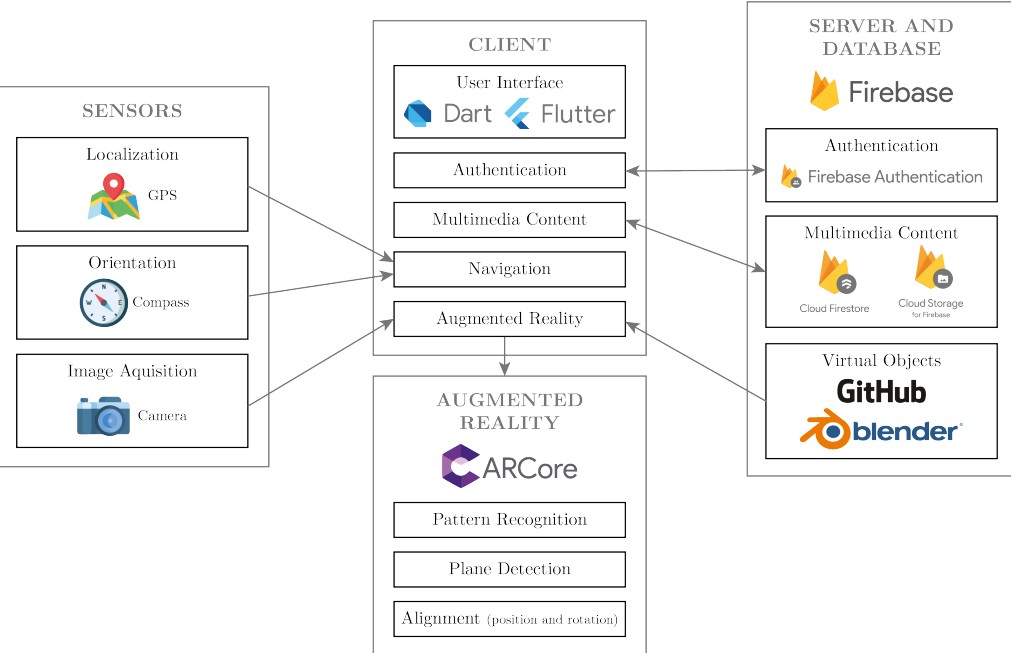

**Figure 9.** Technological architecture of the mobile application.

Figma was the tool used to create the mockups and the non-implemented prototype of the mobile application.

*4.3. Trail Information*

The launching screen (*HomeScreen*) is shown in Figure 10, where the authentication options are provided, including the registration, shown in Figure 11. After that, the users are redirected to the screen shown in Figure 12 (*TrailsInfoScreen*). A trail can be chosen by clicking on the markers on the map.

Selecting a trail allows the user to observe some basic information about it in the sliding panel shown in Figure 13, such as the difficulty, duration, and length of the trail.

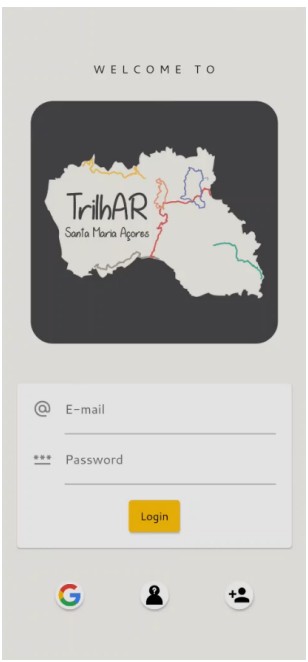

**Figure 10.** User *Login*.

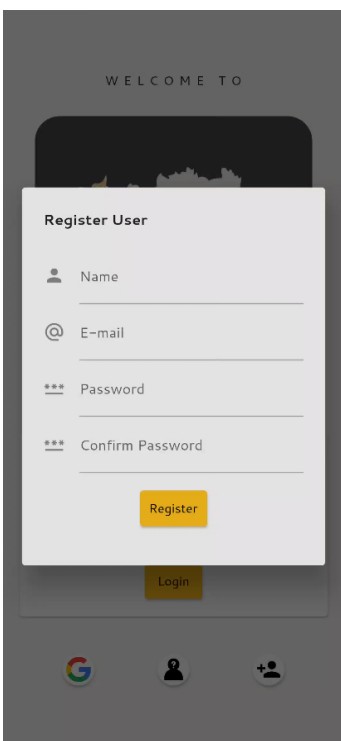

**Figure 11.** Register User.

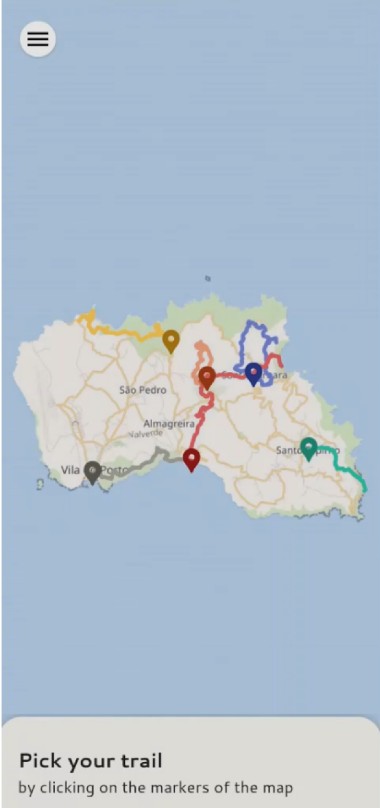

**Figure 12.** After Login.

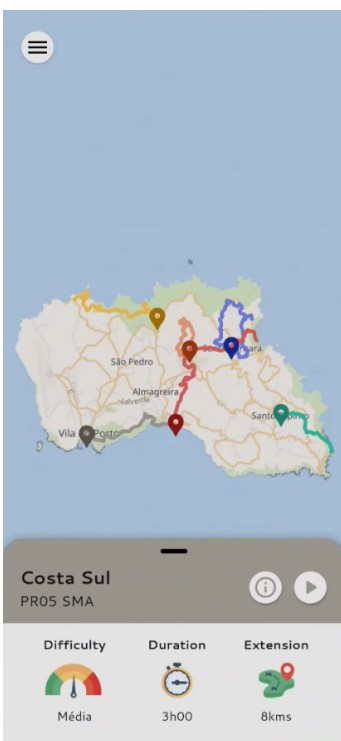

**Figure 13.** Basic Information.

By clicking on the information button ⓘ in Figure 13, the user is redirected to the screen shown in Figure 14 (*TrailInfoFullScreen*). On this screen, more details about the hiking trail are available: map, topography, main points of interest, and images.

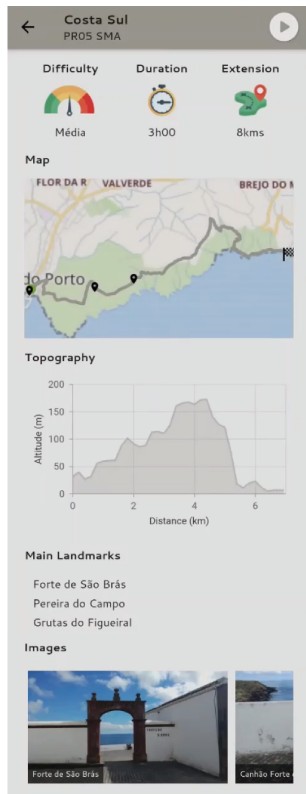

**Figure 14.** Completed Information.

### 4.4. Navigation

To start the hike, the user must click the play button ▶ in Figure 13 or Figure 14. The user is then redirected to a new screen (*TrailNavigationScreen*), where the route is marked as well as the points of interest along the way. It is also possible to observe the distance, the time elapsed, and the current position of the user, allowing the user to understand if he is following the right path (Figure 15). When the trail is completed, there is a possibility to save the completed trail for future reference (Figure 16).

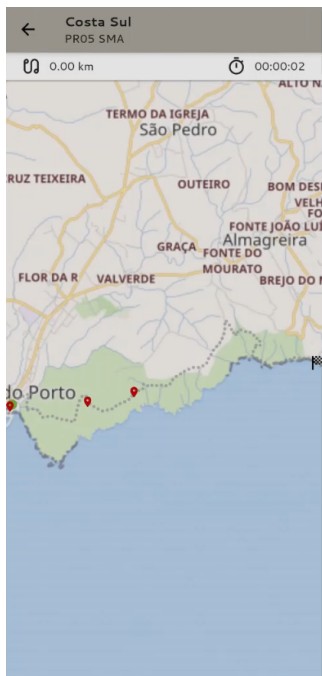

**Figure 15.** Navigation.

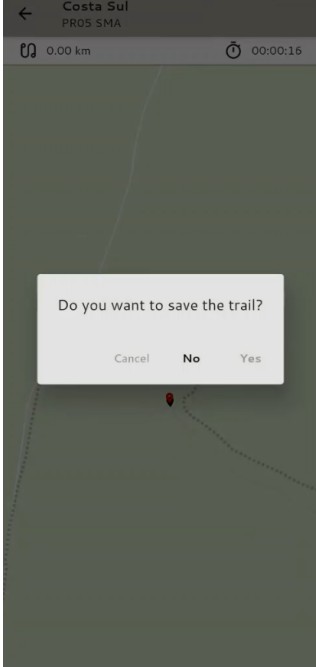

**Figure 16.** Save Trail.

Notifications

To ensure the safety of the visitor, it is important that they stay focused on the hiking trail instead of constantly looking at their smartphone. Thus, a local notification system was implemented to make sure the user is alerted only when they are close to a point of interest ahead (10 m). The diagram in Figure 17 helps explain the triggering of the notification system.

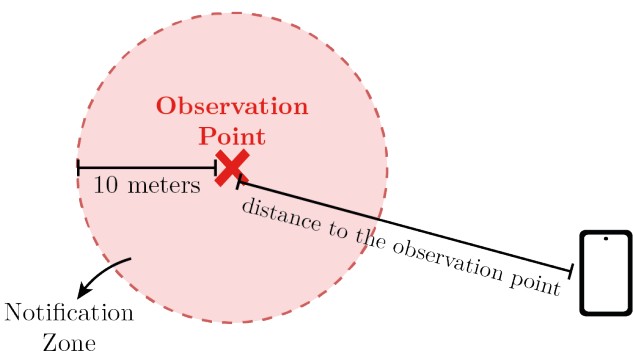

**Figure 17.** Diagram to explain the notification trigger.

*4.5. Augmented Reality*

By clicking on the local notification (Figure 18) or the AR button ⬡, the user is redirected to a new screen (*TrailARScreen*), where the Augmented Reality component is implemented. Two possible views can be presented: Compass View (*COMPASS_VIEW*) and AR View (*AR_VIEW*).

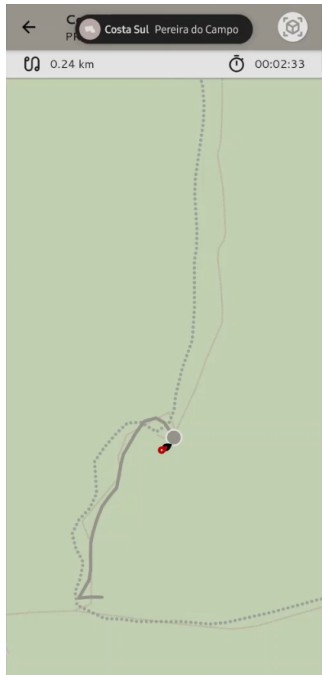

**Figure 18.** Notification.

In the first case (*COMPASS_VIEW*), it is assumed the user is not turned in the direction of the virtual object (the relative bearing is outside of an interval, see Figure 19); therefore, the information on the screen (Figure 20) indicates that the visitor needs to turn around (making sure the red line matches the black one) so that the user is in the right orientation to see the object.

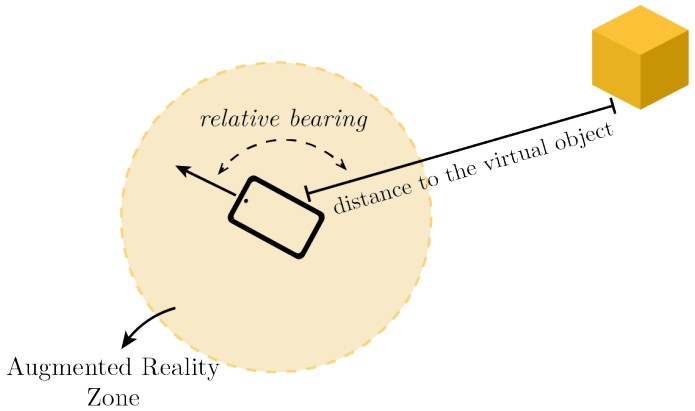

**Figure 19.** Diagram to explain the Augmented Reality system.

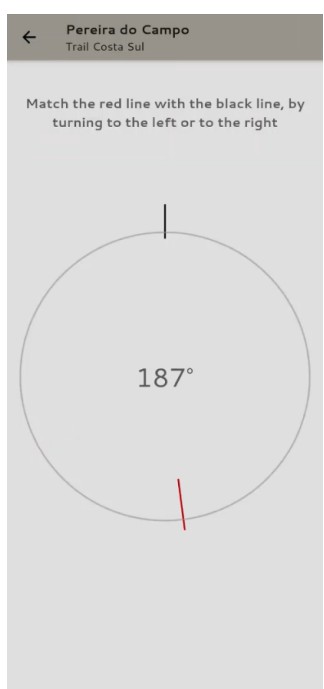

**Figure 20.** *COMPASS_VIEW*.

In the second case (*AR_VIEW*), the user is turned in the right direction, the augmented reality process starts.

The process starts with the Compass View (COMPASS_VIEW) and consulting the smartphone's compass. Thus, it is possible to understand whether the user is headed to the place where the virtual object was previously registered or not. The change from COMPASS_VIEW to AR_VIEW is performed based on the orientation of the device relative to the point of interest. It was considered that from a device orientation of less than $\pm 20^{\circ}$ (340° to 20°), the application should change to AR_VIEW. However, when the application is in AR_VIEW, this range is increased to avoid instabilities caused by the systematic change of views, that is, the range varies according to the view the application is in (hysteresis). Figure 21 shows a diagram that represents the entry conditions in each of the views (taking into account only the orientation of the device).

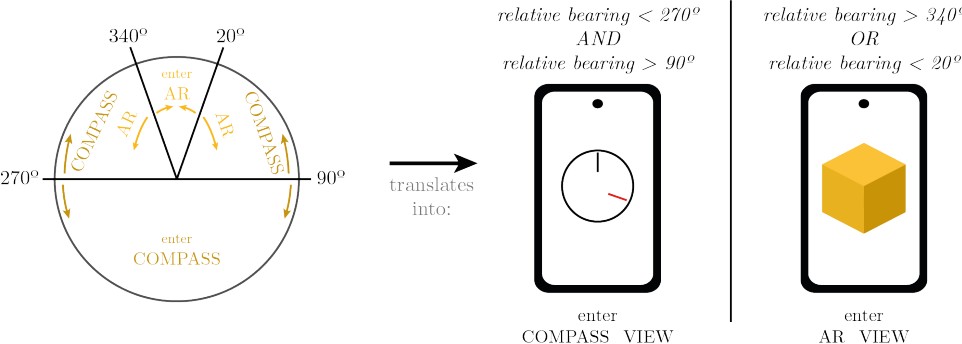

**Figure 21.** Representation of the changes between *COMPASS_VIEW* and *AR_VIEW*.

When the application enters the AR_VIEW, a plane is detected in the environment captured by the smartphone's camera (using ARCore's SLAM) and after that, the rendering of the virtual objects starts.

In order to make sure the object is well aligned with the view of the smartphone's camera, a few calculations (specified in Section 2.1) need to be conducted, regarding the position and rotation of the virtual object itself.

The user is constantly being informed of the process that is happening at the moment, as shown in Figures 22 and 23.

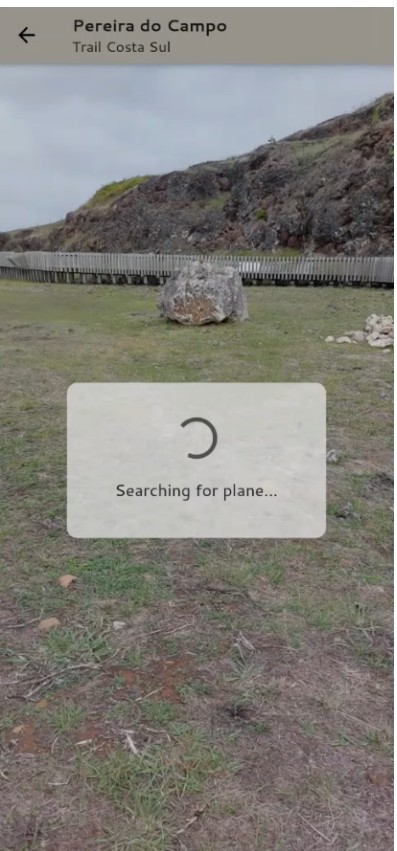

**Figure 22.** "Searching for plane...".

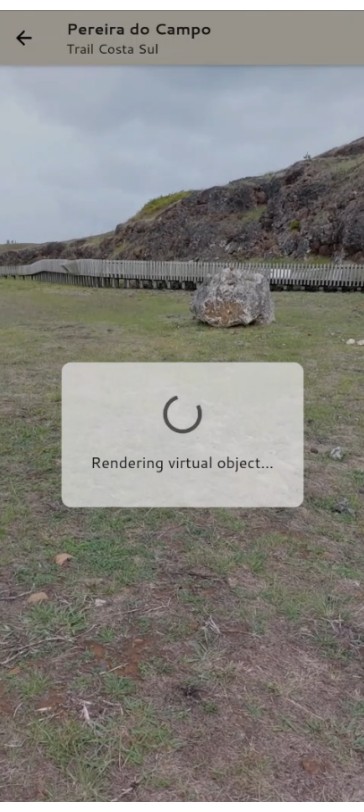

**Figure 23.** "Rendering virtual object...".

Figures 24–26 show examples of the virtual objects present in one of the trails. The virtual object consists of the 3D model of an object allusive to the physical place and a placard with informative and concise text.

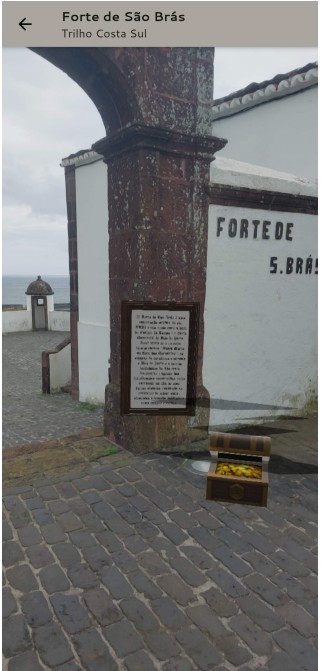

**Figure 24.** Treasure Chest.

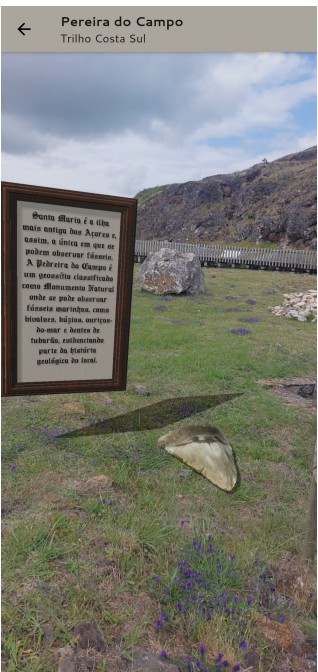

**Figure 25.** Fossil.

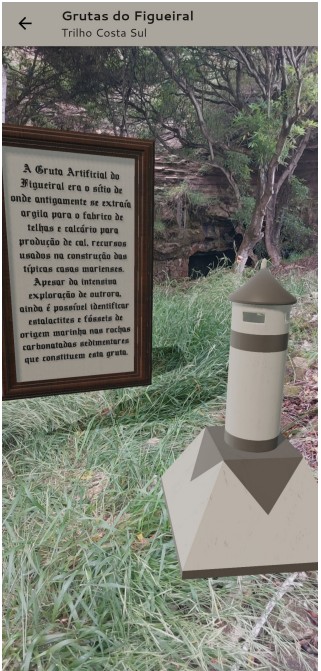

**Figure 26.** Traditional Chimney.

*4.6. Completed Trails*

After finishing a trail and saving it, the user can observe on the respective page (*CompletedTrailsScreen*) the previously completed trails sorted from the most recent to the oldest, with information about the name of the trail, the starting date and time, the time taken to complete the trail and the points of interest visited. It is also possible to delete the trails using the *swipe* movement.

### 4.7. Administrator/Points of Interest

Before having access to all the Augmented Reality experiences provided by the mobile application, the administrator needs to add the points of interest to the respective hiking trails (*AdminScreen*). To perform this, the administrator must go to the respective location and determine the direction in which the object should be placed (by using the corresponding functionality of the proposed application and pointing the smartphone in the desired orientation). It is also necessary to define the name and determine how far the virtual object should be from the current observation point (by inputing the distance manually). It is possible to delete the points of interest by using the *swipe* movement.

### 5. Evaluation and Discussion

The evaluation of the mobile application is mainly focused on the evaluation of the user experience promoted by the interaction process based on the augmented reality component. This also includes the assessment of certain performance criteria that can influence the augmented reality component. The experiments were conducted in a real context with users belonging to the audience on-site, in Santa Maria island, Azores.

The tests were carried out individually by each participant using their own smartphone. The hiking trail chosen for the field tests was *PR05SMA Costa Sul*. The participants were briefed about the goals of the tests and told to follow the questionnaire provided by the supervisor. The supervisor took notes and recorded problems/suggestions explicitly mentioned by the participants.

The questionnaire is divided into six main sections: (1) Application setup, (2) user characterization, (3) before starting the trail, (4) during the trail, (5) after the trail, and (6) overall evaluation of the application. Participants should answer the first two sections before starting to test the application. Sections 3-5 should be followed and answered by participants while testing the application, and Section 6 should be answered after testing the application.

The questionnaire is mainly composed of multiple-choice questions. There are also open questions that allow the participant to write errors, suggestions, or comments. System Usability Scale (SUS) [40] questions and the User Experience Questionnaire (UEQ) [41] are also included in Section 6 of the questionnaire used.

### 5.1. Participants

The tests were performed by 10 participants living on Santa Maria island and mostly with ages between 26 and 40 years old. The first contact of the participants with the mobile application was during the experiment.

Most of the participants use their smartphones from 1 to 5 h a day, mostly to communicate and access social media networks. Only two of the participants frequently use mobile applications for monitoring their hiking trails. Nine in ten participants use Samsung smartphones, with one being Huawei.

### 5.2. Performance

Performance tests were carried out to understand the location system's accuracy and calculate the error inherent to this approach, how the smartphone's battery is influenced by the use of the mobile application, and also the rendering time of the virtual objects.

Most of the performance tests were conducted in a more controlled environment with the smartphone Samsung Galaxy A52s 5G (2021), 256 GB 8 GB RAM, CPU Snapdragon 778 G, a battery of 4500 mAh with an accelerometer, gyroscope, compass, proximity sensor, and camera.

The GPS accuracy was tested in a controlled route inside the campus of the Lisbon School of Engineering (ISEL), as illustrated in Figure 27, where the real position (yellow symbol) and the one indicated in the mobile application location system (red symbol) was marked. By calculating the distance between both locations, an average error of 8 m was obtained. These results were not obtained on Santa Maria Island. This means the average

error can be slightly different. These preliminary experiments were made to verify whether the location error is relevant enough, in terms of the user experience.

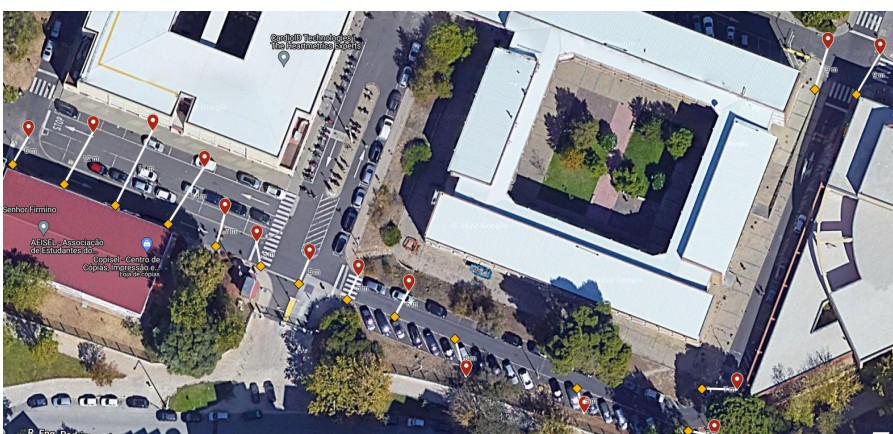

**Figure 27.** GPS Accuracy Tests.

In order to test the battery life with different types of smartphones, users were asked to indicate the battery value at the beginning and end of the hiking route. After analyzing the data obtained, the average battery consumed during the tests was 13.6%. In the graph of Figure 28, it can be observed that, for the most part, older devices have a higher battery drain than newer devices.

Since it is not possible to concretely measure how long each object takes to render for each user, the participants were asked for their opinion on how they felt the waiting time was. By their responses, it can be concluded the users felt that the rendering time was slow for the first object, however, the waiting time for the remaining objects decreased or remained the same.

However, when asked which objects they were able to observe, only 6 of the 10 users were able to visualize the last object on their smartphone (traditional chimney). It is possible to understand that it happened on smartphones with less processing power and RAM. This fact is also due to the decrease and sometimes complete lack of a mobile network in the place where this virtual object was positioned, making it impossible to download the information.

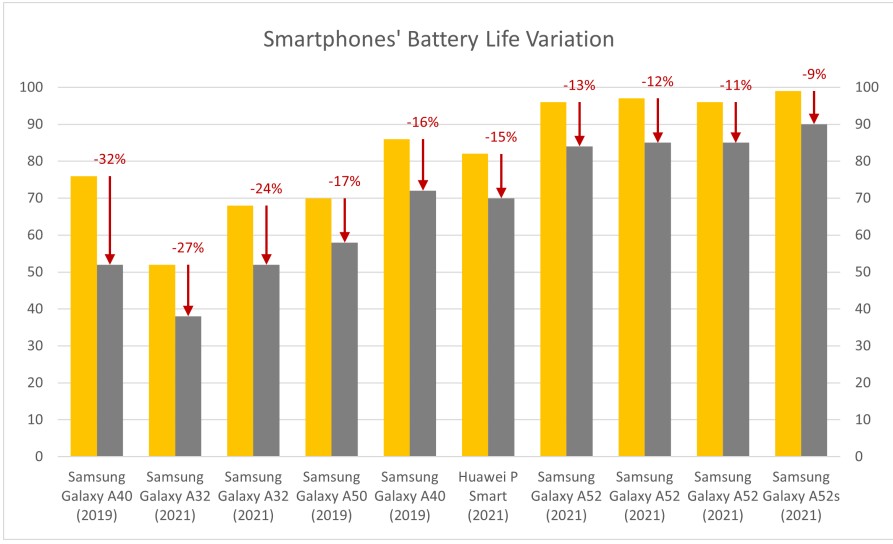

**Figure 28.** Smartphones' Battery Life Variation.

### 5.3. Usability

Section 6 of the questionnaire, besides the SUS, also includes a set of specific questions to measure the usability of the application. When analyzing the answers, in general, the results were very positive, as they considered the application to be simple, intuitive, and aesthetically appealing.

SUS is a standard questionnaire used to evaluate the overall usability of the application. It consists of ten questions with five response options (from "I strongly agree" to "I strongly disagree") and is a very easy-to-perform questionnaire, valid even for small user samples.

After calculating the SUS score of each questionnaire, the average of the results is calculated. Thus, according to the participants, this application has a SUS score of 83, getting a rating of B (Excellent) on the scale used by the SUS [42].

### 5.4. User Experience

UEQ is used to evaluate the user experience, and it consists of 26 pairs of opposite adjectives related to the properties of the application. It is intended to be answered as spontaneously as possible since immediate evaluation is the most important thing for this type of questionnaire. Using this procedure, the mobile application is evaluated according to six parameters: **Attractiveness**, **Perspicuity**, **Efficiency**, **Dependability**, **Stimulation**, and **Novelty**.

The average results for each of the UEQ evaluation parameters (vertical bars), as well as their variance (black lines on top of the vertical bars), can be observed in Figure 29. The parameters where the worst results were obtained are efficiency and dependability. By analyzing the pairs of opposites corresponding to these categories, it can be observed that most users considered the product slow and unpredictable. When questioning the participants that expressed this opinion, it was determined that the application was slow because of the rendering time of the virtual objects and that the notion of the unpredictability considered was not related to the control of the application, but to the fact that the application itself was innovative and unexpected.

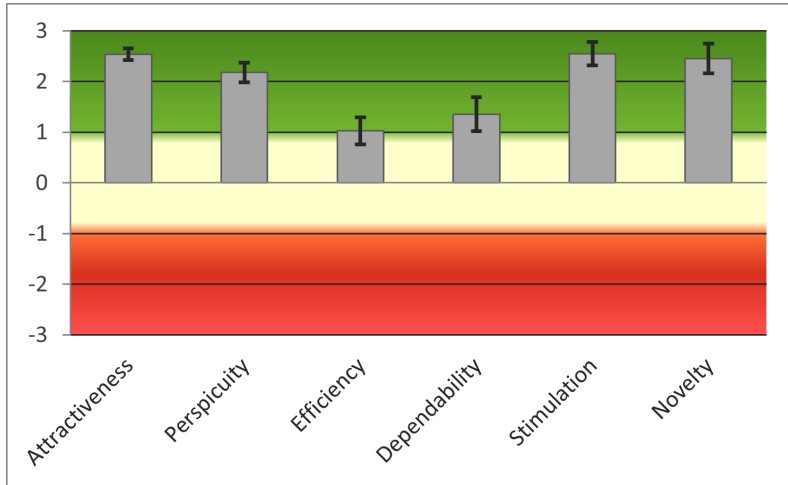

**Figure 29.** Mean and variance values of each parameter.

The results obtained in the evaluation of this mobile application were compared with the benchmark data provided in the Data Analysis Tool [43] (containing information on 468 studies with a total participation of 21,175 people). When comparing the data (Figure 30), it is possible to verify that regarding attractiveness, transparency, stimulation, and innovation, the results were excellent as compared to the top 10%. However, regarding efficiency, the result was below average, and regarding control, the result was above average.

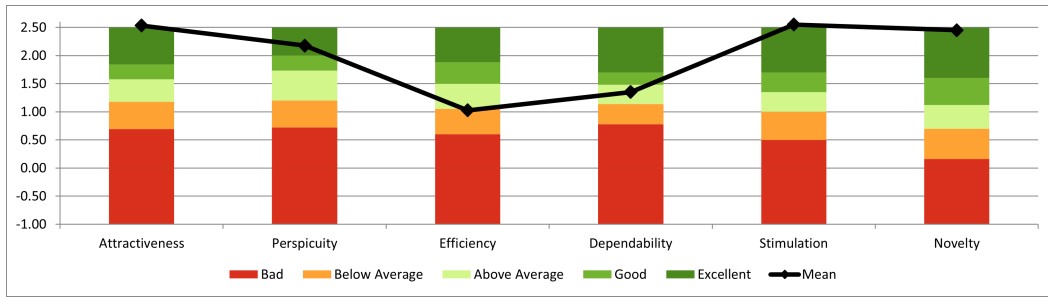

**Figure 30.** Comparison with benchmark data.

### 5.5. User's Opinions

Most users revealed that they had never used Augmented Reality applications on their smartphone before and the integration with the island's hiking trails was very interesting. It also increased their curiosity about the technology itself and its different applications. In addition, they also considered it a good feature to send the notification near the point of interest to draw the attention of visitors. As for the objects presented, most of them liked them, but they liked the object that was modeled especially for this purpose (Figure 31) even more, since it better fits with the place and represents the island of Santa Maria. In this experiment, the application used a set of general 3D objects obtained in the free assets store and a specific 3D object for the Santa Maria Island created in the scope of this work. In the future, it is planned to create more specific 3D models to enrich the visit to the island of Santa Maria.

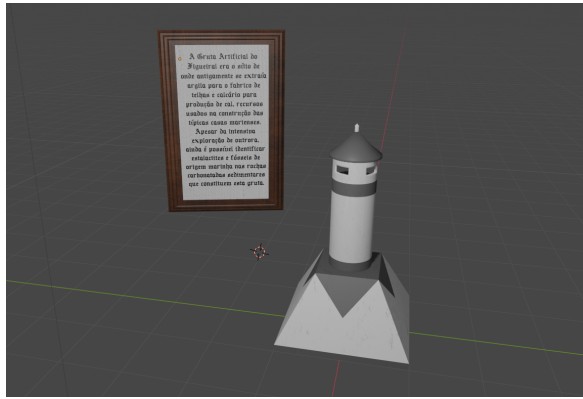

**Figure 31.** Traditional chimney in the modeling software (Blender as stated in Section 4.2).

### 6. Conclusions and Future Work

Tourism on the island of Santa Maria, Azores, has been increasing due to its characteristics in terms of biodiversity and geodiversity. This island has several official hiking trails and the information available can be found on placards or physical and online pamphlets.

Augmented Reality in the tourism sector brings even more interactivity to the already well-known mobile tourist guides. By conducting a study among the target audience, it is possible to understand that their curiosity about Augmented Reality is still low, however, navigation using GPS would be an added value pointed out by most of the respondents.

During the course of this project, it was possible to study and develop a mobile platform to monitor the walking trails of Santa Maria Island, Azores. This provides visitors with a more interactive and recreational experience through Augmented Reality, helps in navigation during the route, and provides updated information and easy maintenance.

The development of the mobile platform using Flutter brings great added value to the continuation of this project since it is a cross-platform technology, that is, the same code can be used in different mobile operating systems. The fact that it is a natively compiled language increases its performance on the smartphone.

The use of Augmented Reality to provide information to users generated great curiosity among the locals who participated in the evaluation tests carried out on the island of Santa Maria. Most of the participants revealed that they had never used AR on their smartphones before and were generally enthusiastic about their experience using the developed mobile application.

Although the proof of concept was successful, this application can always be improved by integrating new features and taking into account the suggestions of the participants in the evaluation. Thus, adding more trails and routes on the island and information (textual and audio on the AR side) about them would be the first step to extending the reach of this application. This information would be added as a new functionality of the administrator role. In addition, the possibility of downloading a priori information about the trail would be an added value in a way that there is no constant need to connect to the internet to acquire the information, since, in some parts of the island, connectivity is not always possible. It could also be a way to reduce the time of downloading and rendering virtual objects. The modeling of virtual objects even more related to the history of the island would further motivate the target audience to observe them through the mobile application.

Through conversation and interaction with some of the participants, it is noticeable that the focus on the mobile application is somewhat lost when the points of interest are too far away. This can be a positive aspect, to the extent that visitors remain focused on the route and the landscape and not on the smartphone, but from the point of view of the experience on the mobile platform, this would be improved if there were points of interest closer together, providing a greater sense of continuity, as well as helping guide the visitors. Therefore, this would be one of the points to be reviewed in the future.

**Author Contributions:** Conceptualization, R.S., R.J. and P.J.; methodology, R.S., R.J. and P.J.; software, R.S.; validation, R.S., R.J. and P.J.; formal analysis, R.S., R.J. and P.J.; resources, R.S.; data curation, R.S.; writing—original draft preparation, R.S.; writing—review and editing, R.S., R.J. and P.J.; supervision, R.J. and P.J. All authors have read and agreed to the published version of the manuscript.

**Funding:** This research received no external funding.

**Informed Consent Statement:** Informed consent was obtained from all subjects involved in the study and the participation to the study was voluntary.

**Data Availability Statement:** Questionnaire data are available upon request from the corresponding author.

**Acknowledgments:** The authors would like to thank all the participants for volunteering their time to conduct the experimental tests and for responding to the survey, as they gave us their time to contribute to our study. This work is supported by NOVA LINCS (UIDB/04516/2020) with FCT.IP.

**Conflicts of Interest:** The authors declare no conflict of interest.

## Abbreviations

The following abbreviations are used in this manuscript:

| | |
|---|---|
| AR | Augmented Reality |
| GNSS | Global Navigation Satellite System |
| NFT | Natural Feature Tracking |
| SLAM | Simultaneous Localization and Mapping |
| RAM | Random Acess Memory |
| CPU | Central Processing Unit |
| SUS | System Usability Scale |
| UEQ | User Experience Questionnaire |
| ISEL | Instituto Superior de Engenharia de Lisboa |

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
