# Peer review of "Development and Evaluation of a Mobile Application with Augmented Reality for Guiding Visitors on Hiking Trails"

_mti, doi:10.3390/mti7060058_

Round 1

Reviewer 1 Report

This paper describes  the design and development of a mobile tour app for guiding visitors in hiking trails on the island of Santa Maria, Azores. It also presents the main issues of the study conducted to evaluate the user experience of the proposed app. The paper is well written and easy to follow. However, some grammar and syntax errors are present in the manuscript. Authors should further revise it to improve the use of English language.

The methodology followed is solid and clear. However, the literature review can be improved and the authors should further elaborate on it. Especially regarding the sub-section “2.2 mobile tour guides” the authors just state a paper that contains an extensive state of the art analysis and do not make any further remarks about it. They  then present a couple of interesting (according to them) works, although more recent works exist. For example, [1] presents an AR social mobile tour guide app that allows many types of interactions between users, and enables curators to create exhibition guides.  Also, [2] presents a mobile app that facilitates better navigation and digital socializing, which enhance user engagement and increase the attractiveness of exploration of trails.

[1] Grammatikopoulou, A., and Grammalidis, N. (2023). Artful—An AR Social Self-Guided Tour App for Cultural Learning in Museum Settings. Information, 14(3), 158.

[2] Yew, J., Deshpande, S., Precians, N., Cheng, K., & Yi-Luen Do, E.(2020). A location-aware app to support the heritage trail experience in Singapore. Journal of Heritage Tourism, 15(6), 680-695.

The presented user research in the next sections is very useful because it gives the audience an idea about the users of the proposed app. Moreover, the architecture and the navigation of the system is thoroughly described. However, some further elaboration explaining why certain choices have been adopted (e.g. Dart/Flutter or Firebase) could be added. Also, the authors should provide some more information about how trail tour guides are developed and added to the mobile app. Is their content static or it can be adapted? To be more precise, are there different user roles for visitors and curators to edit the trail information, or the app has to be edited and published again to update its content? In addition, does the app require a continuous internet connection? And if so, how the app responds when internet connection is lost?

The evaluation methodology is properly described, although the sample of participants is small. However, more importantly, the authors do not present any information how the users were selected to evaluate the proposed app. Moreover, the educational character of the mobile app is not evaluated at all. Users should have been asked about the provided information and how it was communicated to them. Was it interesting and useful? If any such data have already been collected, they could be added to the evaluation description.

The overall evaluation results are quite positive, verifying the attractiveness of the app. However, the participants characterized the proposed tool as not efficient, a fact that possibly has to do with the slow rendering time of virtual objects. Therefore, the authors should further elaborate on the conclusion section and their future work description explaining how they will solve such issues in the future. Moreover, they should compare their work to other recent relative research works and highlight the advantages and disadvantages of their proposed app and how their future work will be influenced according their analysis.

Specific comments:

l. 82: “It was evaluated in a cultural heritage site in a tourism context.” : Please rephrase.

l.309 : “inside” appears twice

Some grammar and syntax errors are present in the manuscript (see attached pdf).  

Author Response

We would like to thank the reviewer for their thorough analysis of the work and for the comments made to help us improve our article. The response to the comments can be found in the attached pdf.

Reviewer 2 Report

The paper discusses a problem related to how to better present tourism information so as to enrich the tourist experience. The development of a mobile application with augmented reality is explained as well as the user opinion is summarized.

I have the following comments:

1). This work tries to say many things in one article, starting from the features of augmented reality, going through mobile application development and its features and reaching to user opinion evaluation. On the other hand, the title emphasizes the evaluation of the user experience. I would recommend the title to be: Development of a Mobile Application with Augmented Reality and User Experience Evaluation (or something else).

2). The purpose of the article should be clearly and precisely stated in the abstract and at the end of the section Introduction.

3). The text on Figure 1 is very small and unreadable. 

4). Section 2. Related work should be titled Background, because explains some concepts used in application development. Also, the authors could add a new section: Related work and to discuss current achievements in development of mobile applications with augmented reality and especially with application in tourism.

5). Figure 3 by the authors created? If not to cite the source.

6). Sub-Section 3.3. Architecture and section 4. Mobile application must be merged with the title: Development of mobile application. 

7). Architecture on Figure 7 I suggest to be Functional architecture, Architecture on Figure 8 - Technological architecture.

8). Conclusion is more general. It must summarize the main findings.

9). Please, check for typos.

The English language requires minor corrections. Please, check for typos.

Author Response

(The authors gave the same response as above.)

Reviewer 3 Report

Summary, this article presents the development and evaluation of an application for mobile devices for guiding visitors in the hiking trails with Augmented Reality (AR). The paper is well structured, and the methodology is clear and adequate. This study has a good background study to support the results. The findings are in line with the results obtained.

In my opinion, a section is missing with the discussion of the results, where the authors should compare the results obtained with other applications and similar studies presented in the literature review.

Author Response

(The authors gave the same response as above.)
